# *Amigos de Fibro* (Fibro Friends): Validation of an Educational Program to Promote Health in Fibromyalgia

**DOI:** 10.3390/ijerph19095297

**Published:** 2022-04-27

**Authors:** Mateus Dias Antunes, Ana Carolina Basso Schmitt, Amélia Pasqual Marques

**Affiliations:** Department of Physiotherapy, Faculty of Medicine, Speech Therapy and Occupational Therapy (FOFITO), University of São Paulo (USP), R. Cipotânea, 51—Vila Butantã, São Paulo 05360-160, Brazil; carolinaschmitt@usp.br (A.C.B.S.); pasqual@usp.br (A.P.M.)

**Keywords:** fibromyalgia, health education, educational technology, health promotion, rheumatology

## Abstract

Health education is one of the main items to enable the promotion of health for individuals with fibromyalgia (FM) in Primary Health Care (PHC) in Brazil. The purpose of this study was to validate a multidisciplinary educational health promotion program called *Amigos de Fibro* (Fibro Friends) for individuals with FM. Methodological research involving 23 health professionals (expert judges) and 45 individuals with FM (target audience) used an instrument to assess the objectives, proposed themes and initiatives, relevance, writing style, and structure of the program through the Delphi technique. The content validity index (CVI) ≥ 0.78 and coefficient kappa ≥ 0.61 were used for data analysis. All 25 items evaluated in both groups presented considerable minimum CVI by CVI and the kappa coefficient. In the global evaluation of *Amigos de Fibro*, the CVI of the specialist judges was 0.90, while the values of the target audience judges were 0.95. The kappa coefficient of the expert judges was 0.90 and that of the target audience judges was 0.85. *Amigos de Fibro*, a light technology in health, was considered with adequate content validity and internal consistency and is, therefore, valid in the use by health professionals with the target audience in PHC, making it possible for them to act as health-promoting agents.

## 1. Introduction

Fibromyalgia (FM) is a rheumatic syndrome, considered a chronic disorder characterized by generalized and persistent non-inflammatory musculoskeletal pain. Concomitant symptoms often include fatigue, insomnia, morning stiffness, depression, anxiety, and cognitive problems (forgetfulness, concentration difficulties, mental sluggishness, memory and attention problems, among others) [1,2].

In primary health care (PHC), there is a proportion of one in 20 patients who seek care and this number is increasing thanks to the increasing frequency of identification of the syndrome and more targeted treatments [3,4,5].

Regarding treatment of FM, the most efficient strategy includes a combination of pharmacological and non-pharmacological interventions. The revised European League Against Rheumatism (EULAR) recommendations for the management of FM indicate that the initial strategy should focus on patient education and non-pharmacological interventions [6].

Thus, pharmacological support is an important complement, while non-pharmacological interventions should be the primary option and focus on the process of adapting and coping with FM in everyday life. For example, physical exercise is beneficial for physical and functional abilities and FM symptoms, while behavioral interventions can improve pain-coping and reduce anxiety, which are the cornerstones of FM treatment management [6].

Health education is one of the main items to enable health promotion in PHC in Brazil and should prepare people to take control of, and responsibility for, their own health and the health of their territory, as well as prepare them for empowerment, decision-making, participation, social control, and acting on the conditioning and determining agents of their health and quality of life [7].

Despite progresses in classifying, treating, and managing symptoms, FM remains an important public health problem today. Thus, it is necessary to increase health education strategies within the scope of PHC, in order to empower individuals about health promotion, self-management of symptoms, and adequate treatment so that they are co-responsible for the knowledge and control of the syndrome [8]. 

The key to effective treatment of individuals with FM in PHC is an integrated and multidisciplinary treatment, with a combination of clinical and non-clinical support, health education, and clarity of goals and expectations. The approximation between the health team and the individual with FM should be part of the intersectoral practices in treatment within the scope of PHC [8].

In this sense, the importance of empowerment for self-care in FM is related to the pedagogical meaning of the term, for health education initiatives, so that individuals with FM acquire knowledge, skills, and are able to recognize possible pain triggers and learn to deal with the adversities of the illness, understanding the signs of their own body during a pain crisis, trying to protect themselves. Thus, with their individual strengths, they can take action and act in search of revising their habits, ways of living, acquiring self-knowledge, self-confidence, and self-control for the maintenance and strengthening of health with total autonomy [9]. According to our search in the scientific literature, there are no detailed programs and/or protocols that promote the health of people with fibromyalgia through health education to promote health, mainly with a participatory and interdisciplinary approach. In addition, there are few studies with good methodological quality on this topic.

Despite all the advances made over the years in the knowledge and management of FM, it remains a challenge for primary care professionals [5]. In this sense, the objective of this study was to validate a multidisciplinary educational health promotion program for individuals with FM.

## 2. Materials and Methods

This is a methodological research that aims to elaborate, evaluate, and validate the technologies developed, in order to ensure their reliability for use in PHC [10]. The projection of our study is described in Figure 1. After the conception of *Amigos de Fibro*, we decided to submit it to evaluation by a group of judges for consultation on future events by means of the technique called “Delphi”, using a questionnaire that can be passed on multiple times until the answers of the judges reach convergence, that is, a consensus that represents the consolidation of the intuitive judgment of the group [11]. 

In this study, the following evaluation rounds were proposed: groups of professionals and individuals with FM who listed their demands through the focus group; evaluation of *Amigos de Fibro*, built with the information and results obtained from the first round, regarding the objectives, proposed themes and initiatives, relevance, writing style, and structure of the program (with specialists and individuals with FM), and final evaluation of the material after the corrections are made, based on the judges’ suggestions [12].

The meetings will be organized by two physiotherapists, who will invite a multidisciplinary team working in primary care to give lectures and hold debates, rounds of conversation, and dynamic meetings. In addition, a physiotherapist will apply the physical exercises described in “*Amigos de Fibro*”. The invited professionals will be a physiotherapist, a physician, a psychologist, a nutritionist, a nurse, a pharmacist, a speech therapist, an occupational therapist, a social worker, and a naturologist.

Each meeting must be structured in at least three basic sections: an initial moment of group preparation for the work of the day (team-building exercises, warm-ups, relaxation); an intermediary, where the group is involved in a variety of activities that facilitate their reflection and elaboration of the developed subject; and a moment of systematization and evaluation of the work of the day, which allows to the participants the visualization of their production as a work group. In addition, this program will be based on the cognitive behavioral therapy model, which is one of the therapies that act on the effects regarding behavior, emotions and symptoms. 

The programming of “Fibro Friends” will be conducted by the invited professionals, according to the themes of each meeting that were developed in the first phase of the study, based on the National Health Promotion Policy, and is presented in Table 1.

The structure of *Amigos de Fibro* was sent to the judges via email, in PDF format. In addition, the questionnaire was made available to them through Google Forms. The expert judges and the target audience (individuals with FM) were not the same as those who participated in the construction of *Amigos de Fibro*.

The participation of experts, here referred to as judges, is indicated in a content validation process. In this process, they must initially evaluate the content as a whole, determining its scope, that is, whether each domain or concept was adequately covered by the itemset and whether all dimensions were included. In this phase, they can suggest the inclusion or the elimination of items. In addition, the judges need to analyze the items individually, verifying their clarity and relevance [13].

A sample calculation was performed to determine the number of judges, obtained through the formula *n* = Za2.P(1 − P)/e2. The stipulated values were Za (confidence level) = 95%, P (ratio of agreement of the judges) = 85%, and (accepted difference from what is expected) = 15% [14,15], which resulted in 22 judges. However, Vianna [16] emphasizes the importance of having an odd number of specialists, in order to avoid a tie of the opinions. After a systematic search, described below, the sample of judges consisted of 23 participants.

The choice of judges in the areas of interest was initially carried out by consulting the Lattes Platform of the National Council for Scientific and Technological Development (CNPq) (www.buscatextual.cnpq.br (accessed on 1 August 2021)). Preferably, doctors were chosen, rather than other researchers, with the understanding that professionals with more years of study and experience have a higher level of excellence. In addition, snowball sampling [17] was also used. 

Seeking to establish parameters for the selection of participants in this part of the study, the system for the classification of judges adapted from Joventino’s proposal was adopted, with selection of those who reach a minimum score of five points. Each judge was e-mailed the informed consent terms and the initial version of *Amigos de Fibro* in portable document format (PDF). After agreeing to participate in the study and analysis of *Amigos de Fibro*, the participants accessed the program’s electronic evaluation questionnaire for specialists and sent their answers back to the researcher. 

To complement the validation process, it is recommended that lay individuals be included [18] and that representative individuals of this population be involved [13]. It is recommended that the portion of the target population that evaluates *Amigos de Fibro* be 30 to 40 individuals [19]. In this study, through e-mail, 45 individuals were evaluated (importance placed on the odd number) with a medical diagnosis of FM.

The inclusion criteria considered were individuals of both sexes, aged 18 years or over and with a medical diagnosis of FM, confirmed by the International Statistical Classification of Diseases and Related Health Problems (ICD-10), with the code M79.7, and confirmed by an assessor against the ACR Classification Criteria, 2016 revised version [20].

Exclusion criteria were individuals undergoing physical therapy treatment or who treated in the three months prior; diagnosis of other conditions causing chronic pain (neuropathies, rheumatoid arthritis, osteoarthritis, spinal stenosis, or neoplasia); medically proven severe mental disorders (schizophrenia, psychosis, bipolar affective disorder, severe depression); visual or hearing impairment. All these criteria were considered self-reported by the participants.

Individuals with FM were invited to participate in the study through the Brazilian Association of Fibromyalgics. After the invitation, interested parties received the informed consent terms and the initial version of *Amigos de Fibro* in portable document format. In addition, they also received the program’s electronic evaluation questionnaire for the target audience and sent their answers back to the researchers. Participants were also encouraged to suggest addition or deletion of items [13].

An electronic program evaluation questionnaire was developed by the authors for specialists, as well as an electronic program evaluation questionnaire for the target audience for the group of individuals with FM. The first part consisted of information on socioeconomic profile; the second part presents items regarding the domains (objectives, proposed themes and initiatives, relevance, writing style, and program structure) [21], where a minimum agreement of 75% is required for positive response and, in the third part, an open field is available for participants to express their personal opinions [22,23].

The instrument elaborated for the judges’ and target audience’s evaluations consisted of five evaluation items (objective, proposed themes and initiatives, relevance, style of writing, and structure of the program) with answers based on the Likert scale, with four levels (totally inadequate, partially inadequate, adequate, totally adequate) and fields for general comments and suggestions. The Likert scale is a widely used rating scale, in which the respondent’s attitude is measured over a continuum from highly favorable to highly unfavorable, with an equal number of positive and negative possibilities for responses and a medium or neutral category. For the entire instrument, or for each domain, the score is calculated by adding the participant’s answers divided by the number of corresponding questions [24]. 

After evaluating the instruments, content validation was performed. Content assessment is an essential step in the development of new measures because it represents the beginning of mechanisms to associate abstract concepts with observable and measurable indicators [13]. Content validation is a two-step process: the development phase and the judgment quantification phase. The first requires a literature review to identify the instrument’s content, while the second occurs when judges evaluate the instrument and its items according to the importance of the content domain. In addition, criteria such as clarity and overall completeness of the instrument [25,26] are also evaluated.

Content validation was initially carried out by applying the kappa coefficient to measure the level of agreement and level of consistency (reliability) of the judges in relation to the permanence of the items that make up the instruments [26]. The kappa coefficient is an adjusted agreement indicator that ranges from “minus 1” to “plus 1”: the closer to 1, the better the level of agreement between observers; their distribution and respective levels of interpretation are classified from poor to perfect. For this analysis, answers “1” (totally inadequate) were grouped with “2” (partially adequate). Likewise, answers “3” (adequate) and “4” (totally adequate) were grouped. As an acceptance criterion, an agreement greater than 0.61 was established among the judges [26].

The content validity index (CVI) was also used, which represents a method widely used in the field of health to calculate the level of consensus regarding the representativeness of the items in relation to the content of the study [10,27], since it measures the proportion of individuals who are in agreement on certain aspects of an instrument and its items [13]. For its calculation, the number of judges who rated the item as adequate/totally adequate was divided by the total number of judges (assessment per item), resulting in the proportion of judges who judged the item to be valid. Authors argue that, in the process of evaluating individual items, the number of judges should be considered. With five or fewer subjects participating, all must agree to be representative. In the case of six or more, a rate not inferior to 0.78 is recommended [10,13,25].

The CVI measures the ratio of judges in agreement of a determined evaluated aspect. This method uses the Likert scale with scores from one to four. The index is calculated through the sum of agreement of the items marked as “3” (adequate) and “4” (totally adequate) by the specialists, divided by the total of answers [13]. In this sense, the CVI was calculated for each of the items listed in the blocks and for the total set of items in the instrument (sum of all CVI calculated separately, divided by the number of items), according to the recommendations of Lynn [25], using the cut-off point of 0.78 (78%) for both calculations. The formula to calculate the CVI used was
$$\mathrm{Content}\;\mathrm{Validity}\;\mathrm{Index}\;=\frac{\mathrm{Number}\;\mathrm{of}\;\mathrm{responses}\;3\;\mathrm{or}\;4}{\mathrm{Total}\;\mathrm{number}\;\mathrm{of}\;\mathrm{responses}}$$

After the suggestions made by specialist judges and target audience specialists, *Amigos de Fibro* was adapted, incorporating suggestions in order to meet the proposed needs and expectations. After analyzing the data, it was verified that more than one round of agreement in both groups (expert judges and target audience) would not be necessary, as proposed by the Delphi methodology, since this group reached the CVI above 0.78 for all items, and also global, in the first round [25]. The data were organized in an electronic data sheet in Microsoft Excel and exported to the Statistical Package for the Social Sciences (SPSS) statistical software version 22.0, serial number 10101151049. After coding and tabulation, data were analyzed using descriptive statistics.

This study followed the precepts established by Resolution 466/12 of the Ministry of Health and was submitted for consideration by the Research Ethics Committee of the Faculty of Medicine of the University of São Paulo and approved under opinion no. 3.197.778.

## 3. Results

Regarding the expert judges, the majority (39.2%) were aged between 31 and 40 years, with an academic doctorate as the highest degree (39.2%), and all (100%) were teaching. Practical activity in the area of interest was higher (47.8%) between 1 and 10 years (Table 2).

Regarding the process of judging the items that make up *Amigos de Fibro* by the expert judges, none of them were evaluated as totally inadequate. According to Table 3, of the total 25 items evaluated, all items (100%) had a considerable minimum agreement index by the CVI and the kappa coefficient. In the global assessment by the expert judges, *Amigos de Fibro* obtained a CVI of 0.90 and a kappa coefficient of 0.90.

As for the target audience judges, their distribution across the 27 federative units (26 states and one federal district) was 59.2%, highlighting a greater number of target audience judges in the states of São Paulo (37%) and Maranhão (29.6%).

The majority (91%) of the target audience judges were female, aged between 31 and 40 years (38%), married or living with a partner (65%), and with incomplete higher education (49%). Regarding their self-perception of health, 40% of the target-audience judges considered it bad and administered three or more medications (67%) (Table 4).

Regarding the process of judging the items that make up *Amigos de Fibro* by the target audience judges, none of them were evaluated as totally inadequate. As shown in Table 5, of the total 25 items evaluated, all (100%) had an agreement index greater than the cut-off score in both evaluations. In the global evaluation by the target audience judges, *Amigos de Fibro* obtained a CVI of 0.95 and a kappa coefficient of 0.85.

## 4. Discussion

Before a possible expansion of *Amigos de Fibro* in Brazil, the aim was to validate the program in terms of its content and structure. This means to identify how fully its items are understandable for specialists (professionals) and the target audience (individuals with FM) [21,28]. This fact is essential, as it facilitates its appropriation and reproduction as a health intervention technology and contributes to its scalability and applicability in other municipalities.

In Brazil, some initiatives and/or programs aimed at health education in FM with a focus on reducing the symptoms of the syndrome have been described. However, the literature points out that these initiatives are, for the most part, punctual, restricted to specific municipalities, and carried out by disciplinary characteristics [29,30,31,32,33,34,35]. Thus, the need to build a multidisciplinary program that is easy to reproduce and incorporate in PHC in the country stands out [7,36].

Therefore, the validation of *Amigos de Fibro* was performed by the judging panel composed of professionals and individuals with FM. Among the group of experts (professionals), high agreement occurred in the first round of judgment, as the global CVI surpassed the minimum agreement value established as a program validation criterion (both for the individual items and in global form).

In relation to the target audience group (individuals with FM), all items also reached the minimum value of global agreement in the first round, making it unnecessary to carry out a second round of consensus, as it was higher than previously established. There was also a consensus between the judges of both groups on all items individually evaluated.

It is understood that knowledge in health is a fundamental requirement for self-care, which depends on the empowerment of the individual, who needs to know the necessary actions to prevent a particular disease, considering it important to remain aware and motivated to acquire new habits of life. Therefore, it is emphasized that knowledge is important, but not necessarily a predictor of self-care [37].

A study developed by Moretti et al. [38], which aimed to analyze the level of knowledge about FM in a sample including patients, family members, and professionals from several states in Brazil, found that many participants pointed out the lack of dissemination and information about the syndrome, as well as the lack of access to alternative therapies, better-qualified professionals, and universal and decentralized forms of action. In this sense, *Amigos de Fibro* proposes to meet this demand.

In the context of information on FM, there is a deficit in patients’ knowledge about its repercussions for self-care, thus demonstrating the need for new technologies that are effective in raising awareness and promoting the improvement of knowledge, attitude, and practice of individuals with FM regarding the syndrome, promoting their empowerment and enabling them to change their perception of health and adopt health promotion practices, contributing to greater control of the symptoms [9,29,39,40].

Educational technologies are creative, reliable, and useful tools for knowledge, communication, and health education, directly contributing to the improvement of the teaching–learning process and encouraging healthy practices. It is considered that the ability to perform self-care depends, in part, on the ability to read and understand health information. In this context, written information has been used as a complementary strategy for health education [41].

Many individuals with FM complain of widespread pain, but the underlying cause or mechanism is unknown. Patients who do not understand pain mechanisms seem to describe pain as more threatening or dangerous due to damage or injury [42]. This perception results in lower pain tolerance, more destructive thoughts about pain, and fewer pain-management strategies [42]. Therefore, patient education strategies that provide up-to-date information on FM pain and pathophysiology need to be integrated into the treatment [43].

Empirical evidence has shown the promising effectiveness of pain neuroscience education (END) as a recent method of patient education that allows patients with any chronic pain condition to understand and/or reconceptualize pain, thereby changing their negative beliefs and unfavorable attitudes around pain [7].

In END, participating individuals are deeply informed about the neurophysiology of pain from a neuroscience perspective. Systematic reviews and meta-analyses have shown that PND has a low-to-moderate effect in reducing pain, disability, and psychological distress, also improving physical function, beliefs, and attitudes of patients with chronic musculoskeletal pain, and it has a limited number of health professionals participating in this intervention [43,44,45,46].

Educational follow-up studies with individuals with FM are scarce, and health care that should be planned as multi- and interdisciplinary care has been carried out in isolated form by each professional [7,36,47,48]. The scientific literature presents some studies with educational approaches for individuals with FM. However, the diversity of professionals has always been limited, focusing care on the doctor, nurse, physiotherapist, and psychologist [7,33]. Some programs were only applied by approximately one to three previously trained health professionals [48,49,50], unlike *Amigos de Fibro*, which proposes the use of a range of professionals.

Highlighting the importance of a more complete team, a study by Miró et al. [51] involved professionals of different areas to work with groups of individuals with FM and proved to be promoters of pain relief, generating well-being, and contributing significantly to their quality of life. The treatment of fibromyalgia patients should follow multifactorial models; this will allow a systematic development of the necessary abilities for the transition from rehabilitation to maintaining an active and independent lifestyle [7,33,49,52].

Studies on the effects of health education and social support on FM care began to be discussed from studies carried out in 2001 at the international level, for example, in the study by Oliver et al. [53], carried out in San Diego (United States) on individuals with FM with an intervention that presented educational and social support activities to improve symptoms and health expenses with the syndrome. The results began to show promise. The differences, however, were not significant.

In the following years, new studies were developed with educational interventions to improve the clinical condition, such as the pilot study carried out in Pontevedra (Spain) by Ayan et al. [54], in which there was a combination of health education with physical exercises. It showed that, after the intervention, there was significant improvement in the physical condition of the patients, as well as the impact of FM on the lives of patients.

Other studies began to show positive effects of interventions based on health education, such as the study by Van Oosterwijck et al. [55] in Ixelles (Belgium), which indicated that this approach appeared to be a useful component in the treatment of patients with FM, as it showed improved health status and long-term inhibition of endogenous pain. However, the improvements were minor, and several limitations persisted.

In Brazil, the first studies of a health education program in FM began with the Interrelational School of Fibromyalgia program, which was proposed by Souza et al. [56] and sought to solve flaws identified in other multidisciplinary programs for the treatment of chronic conditions, such as the high dropout rate and high cost of human material and infrastructure. This program was developed specifically for the treatment of people with FM and the meetings sought to teach self-treatment techniques and coping strategies to individuals with FM [56,57]. However, this program is based on the interrelational model proposed by the theories of communication, unlike the previously proposed treatments, and is based on cognitive behavioral therapy [30,56].

In this sense, studies began to be carried out, such as, for example, the study by Pernambuco et al. [32], which verified the impacts on the immune and/or neuroendocrine systems, as well as on the clinical condition of FM, through health education intervention and, thereafter, results suggested that educational intervention could induce subjective and objective changes (immunological and neuroendocrine), which could explain, at least in part, the improvement in the health status of the individual with FM.

At the global level, a systematic review carried out by García-Ríos et al. [33] highlighted that patient education is considered the first step in the self-care of a patient with FM, but the scientific evidence supporting the effectiveness of education in reducing the main symptoms is limited. Future research projected into stronger and more homogeneous intervention is needed [33].

Additionally, Antunes et al. [7] highlighted, in a systematic review, that a health education program, especially in groups, can improve pain and quality of life in people with FM, but it also reinforces that the evidence shows low methodological quality [7]. *Amigos de Fibro* comes with this intention to develop an educative group to promote the health of individuals with FM and to verify its benefits by means of a study of good methodological quality.

Group health education programs for FM around the world, such as *Amigos de Fibro*, have shown that there is a high dropout rate in individual treatments, although when individuals with FM participate in a group, their satisfaction is high [58]. Another important point is the social support that is promoted when carrying out group activities around FM [59,60].

Social support, both face-to-face [59,60,61] and online [62,63], is a coping resource in chronic diseases such as FM. It represents one of the most important factors in health promotion. Social support represents an external resource that is obtained from other people and is operationalized as a social resource. Social support is considered an essential aspect of life in general and mental health and can be defined as a sub-concept of social networks [61,64].

The simple identification of similar cases or symptoms among individuals with similar problems in discussion groups can be a great tool to manage chronic conditions, including FM, offering improvements to the health of users, promoting greater autonomy and proactivity, in addition to benefits such as improvements to social life, reduction of hopelessness, more knowledge about the disease, expansion of behavioral strategies and better clinical results, especially in the areas of rheumatic diseases and syndromes [63,65].

In addition, the proposal of group programs can help in places where there are limited resources available for health departments, where it is not possible to provide personalized self-management programs. Many units in recent years offer self-management programs for groups of FM patients, and it has been a trend in care [66].

Making use of educational initiatives in the treatment of the FM is a valuable resource for pain control, fatigue, and psychological symptoms, with benefits that generally last in the long term, having shown superiority over conventional individual treatment, and with indications of cost-effectiveness in economic analyses [63,67].

Finally, a health education program model, such as *Amigos de Fibro*, has been standing out, mainly because it is a low-cost, non-invasive, very comprehensive method and possibly capable of producing clinical improvements, reducing public and private expenses resulting from FM, reducing patient anxiety, and encouraging them to seek and achieve therapeutic success [56,57].

The initiative is not intended to replace the presence of the physical therapist in the specialized care for individuals with FM, but to offer continuity of care to a group of individuals, as well as to present several relevant issues about the treatment of FM. Thus, the activities developed by this method are considered relevant as a way of promoting health education with greater autonomy of the participants.

*Amigos de Fibro* will contribute to the health system with qualified information that will support the non-pharmacological management of FM treatment in PHC. For the next steps of *Amigos de Fibro*, the research group responsible is conducting an important randomized clinical trial to show the effectiveness of this program in PHC in Brazil. The protocol was registered in the Brazilian Registry of Clinical Trials (ReBEC) under the reference number RBR-3rh759. Thus, researchers will soon be able to provide new, high-quality information about *Amigos de Fibro*.

*Amigos de Fibro* is the first multidisciplinary educational program in Brazil to promote the health of people with FM that was created based on scientific evidence, the clinical experience of different professionals, and the opinion of individuals with FM. The authors suggest that health professionals recommend the use of this program to their patients as an auxiliary resource in promoting health in FM.

Regarding the limitations of the study, it is highlighted that comments and suggestions constitute a rich source of information for the improvement of the program. However, not all items were commented on by the judges, since this is optional in the Delphi methodology. Future evaluations of *Amigos de Fibro*, already modified from the contributions of the present study, may benefit from other qualitative approaches, such as the association of the Delphi technique with focus groups. An inherent bias in the questionnaires is the tendency toward favorable responses; therefore, it is important to be clear that this may be a limitation of the study.

Due to the moment of social isolation that occurred during this study due to the COVID-19 pandemic, participants were recruited from different regions of the country, making it impossible to collect data in person from just one specific location. COVID-19 has fundamentally changed everyday life, patient care, and health research. Numerous trials that were underway prior to the COVID-19 pandemic faced unavoidable modifications in response to the pandemic, such as changes to methods of recruitment, intervention delivery, outcome assessment (e.g., substituting virtual visits for in-person ones), statistical analysis, and, sometimes, study design [68].

Nevertheless, technology has the potential to contribute to the practice of health professionals, as this constitutes an educational program that goes beyond the practice of vertical and traditional health education. It is understood that the structuring of light technologies for health aimed at expanding care practices for individuals with FM is of great contribution to health promotion from the PHC perspective. This argument corroborates the national and international guidelines for the treatment of FM, emphasizing the need to invest in programs focused on education and health that guarantee minimum conditions to empower patients to promote self-care.

Also noteworthy is the potential of the technology in question to qualify the action of health professionals who work in PHC, in the sense of strengthening potential therapeutic support for individuals with FM. In this context, *Amigos de Fibro* has the potential to promote good results, impacting the participants’ quality of life.

### Practical Implications

*Amigos de Fibro* was created from the joint initiative of individuals with FM and health professionals, and can be an effective educational tool to be implemented in basic health units in Brazil, with the aim of promoting self-care, quality of life, and the health of individuals with FM. As few studies on similar studies have been reported in the scientific literature, this study is essential, as it is believed to contribute to the advancement of care for people with fibromyalgia. *Amigos de Fibro* represents an excellent model of light technology in health to promote the education and counseling of individuals with FM.

## 5. Conclusions

After this study, it is considered that *Amigos de Fibro* was considered valid by both the expert judges and the public, since it obtained a CVI above the expected (global and assessment by item). The knowledge resulting from this study showed that *Amigos de Fibro* constitutes a light technology of innovative intervention that can be used in PHC to offer qualified assistance to individuals with FM. The innovation of *Amigos de Fibro* is established in the focus of its initiative: the strengthening of health education, the subject of its intervention (individuals with FM), and the results of its initiative (promotion of self-care). Thus, greater encouragement is suggested for the implementation of educational policies as alternative and/or complementary ways of treating FM, due to the chronicity of this clinical condition and the notorious difficulty in accessing therapeutic care in our universal health system.

## Figures and Tables

**Figure 1 ijerph-19-05297-f001:**
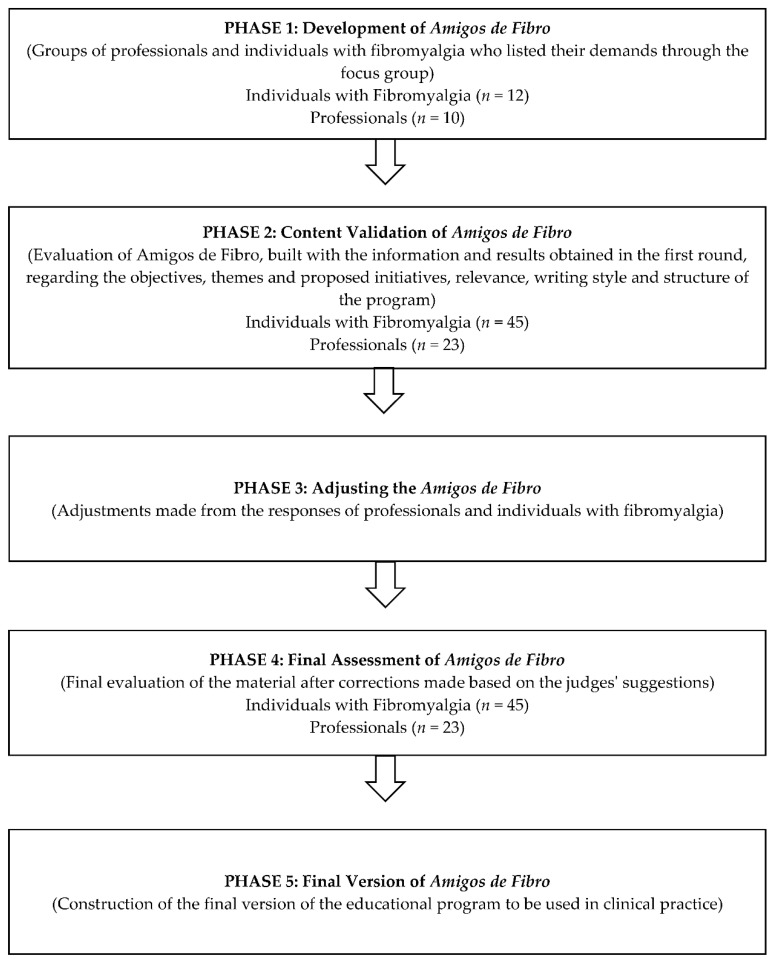
Flowchart of the “*Amigos de Fibro*” validation process.

**Table 1 ijerph-19-05297-t001:** Script of “*Amigos de Fibro*”.

Meeting	Theme	Professional	Physical Exercise Practice
1	Program introduction and socializing	Physiotherapist (educational activity) + Physiotherapist (physical exercises)	The physiotherapist will explain how physical exercises, guidance and practice care will work.
2	Knowing fibromyalgia	Doctor (educational activity) + Physiotherapist (physical exercises)	Muscle stretching exercise
3	Health production and care	Nurse (educational activity) + Physiotherapist (physical exercises)	Muscle strengthening exercises
4	Family and work	Social Worker (educational activity) + Physiotherapist (physical exercises)	Muscle stretching exercise
5	Body practices and physical activity	Physiotherapist (educational activity) + Physiotherapist (physical exercises)	Muscle strengthening exercises
6	Adequate and healthy eating	Nutritionist (educational activity) + Physiotherapist (physical exercises)	Muscle stretching exercise
7	Health and Well-Being	Psychologist (educational activity) + Physiotherapist (physical exercises)	Muscle strengthening exercises
8	Pharmacological approach	Pharmacist (educational activity) + Physiotherapist (physical exercises)	Muscle stretching exercise
9	Integration of the activities	Physiotherapist (educational activity) + Physiotherapist (physical exercises)	Guidance to perform stretching and muscle strengthening exercises at home
10	Integrative and complementary practices	Naturologist (educational activity) + Physiotherapist (physical exercises)	Muscle strengthening exercises
11	Integration of the activities	Physiotherapist (educational activity) + Physiotherapist (physical exercises)	Guidance to perform stretching and muscle strengthening exercises at home
12	Occupational performance	Occupational Therapist (educational activity) + Physiotherapist (physical exercises)	Muscle stretching exercise
13	Integration of the activities	Physiotherapist (educational activity) + Physiotherapist (physical exercises)	Guidance to perform stretching and muscle strengthening exercises at home
14	Sleep Quality	Speech therapist (educational activity) + Physiotherapist (physical exercises)	Muscle strengthening exercises
15	Retrospective	Physiotherapist (educational activity) + Physiotherapist (physical exercises)	The physiotherapist will reinforce the guidelines on the continuity of the practice of physical exercise and its benefits

**Table 2 ijerph-19-05297-t002:** Profile of expert judges (*n* = 23).

Variables	*n*	%
Sex		
Male	10	43.5
Female	13	56.5
Age		
20 to 30 years	4	17.4
31 to 40 years	9	39.2
41 to 50 years	4	17.4
51 years or more	6	26.1
Marital Status		
Married or living with a partner	12	53
Divorced	1	4.3
Separated	-	-
Widowed	1	4.3
Single	9	39.2
Race/color		
White	19	82.7
Brown	3	13
Black	1	4.3
Yellow	-	-
Indigenous	-	-
Profession		
Physiotherapist	4	17.4
Rheumatologist doctor	3	13
Nurse	2	8.7
Psychologist	2	8.7
Nutritionist	2	8.7
Occupational therapist	2	8.7
Speech therapist	2	8.7
Pharmacist	2	8.7
Naturologist	2	8.7
Social worker	2	8.7
Currently Teaching		
Yes	23	100
No	-	-
Years as a Graduate		
1 to 10 years	7	30.4
11 to 20 years	9	39.2
More than 21 years	7	30.4
Highest Degree		
Specialization/Residency	3	13
Master’s Degree	6	26.1
Doctorate Degree	9	39.2
Postdoctorate Degree	5	21.7
Has a course/thesis/dissertation work in the area of interest		
Yes	23	100
No	-	-
Has, in the last five years, published an article on the area of interest in an indexed journal		
1 to 10 publications	16	69.6
11 to 20 publications	3	13
More than 21 publications	4	17.4
Teaching experience in the area		
1 to 10 years	15	65.2
11 to 20 years	6	26.1
More than 21 years	2	8.7
Practical work in the area of interest		
1 to 10 years	11	47.8
11 to 20 years	9	39.2
More than 21 years	3	13

**Table 3 ijerph-19-05297-t003:** Judgments of expert judges (*n* = 23) on *Amigos de Fibro*.

Items	Judgment									
	Totally Inadequate		Partially Inadequate		Adequate		Totally Adequate		CVI *	Kappa
	*n*	%	*n*	%	*n*	%	*n*	%		
1. Objectives										
1.1 Care innovation potential for people with fibromyalgia	-	-	-	-	3	13	20	87	1.00	1.00
1.2 Clarity of the program’s objective	-	-	2	8.7	2	8.7	19	82.6	0.91	0.86
1.3 Relevance of the theoretical framework	-	-	2	8.7	3	13	18	78.3	0.91	0.86
1.4 Basic premises capable of guiding people with fibromyalgia in promoting their health	-	-	-	-	4	17.4	19	82.6	1.00	1.00
2. Proposed Themes and Initiatives										
2.1 The themes and initiatives listed offer support in promoting the health of people with fibromyalgia	-	-	-	-	3	13	20	87	1.00	1.00
2.2 The themes and initiatives listed are important to be applied to people with fibromyalgia in primary health care	-	-	1	4.3	4	17.4	18	78.3	0.95	0.93
3. Relevance										
3.1 Covers relevant topics for in promoting health for people with fibromyalgia	-	-	2	8.7	2	8.7	19	83	0.91	0.86
3.2 Addresses important premises for the practice of health education	-	-	-	-	4	17.4	19	82.6	1.00	1.00
3.3 Is innovative technology	-	-	1	4.3	4	17.4	18	78.3	0.95	0.93
3.4 Is a combination of multidisciplinary knowledge and practices	-	-	-	-	4	17.4	19	82.6	1.00	1.00
3.5 Has scalability potential	-	-	4	17.4	2	8.7	17	73.9	0.82	0.74
3.6 Is an approach that integrates multidisciplinary knowledge	-	-	2	8.7	2	8.7	19	82.6	0.91	0.86
4. Writing Style										
4.1 Ordering of content and initiatives	-	-	1	4.3	4	17.4	18	78.3	0.95	0.93
4.2 Clarity of the text	-	-	3	13	3	13	17	73.9	0.86	0.79
4.3 The vocabulary used	-	-	2	8.7	2	8.7	19	82.6	0.91	0.86
4.4 The extent of the material	-	-	2	8.7	5	21.7	16	69.6	0.91	0.86
4.5 Complexity of the text	-	-	4	17.4	3	13	16	69.6	0.82	0.74
4.6 Common vocabulary for professionals	-	-	2	8.7	4	17.4	17	73.9	0.91	0.86
5. Structure of the Program										
5.1 Number of participating professionals	-	-	-	-	-	-	23	100	1.00	1.00
5.2 Structure of the program (number, frequency, time/duration, period of the day, day of the week)	-	-	2	8.7	4	17.4	17	73.9	0.91	0.86
5.3 Conditions offered by the professional	-	-	1	4.3	2	8.7	20	87	0.95	0.93
5.4 Activities performed by the participant	-	-	1	4.3	3	13	19	82.6	0.95	0.93
5.5 Product of the participant’s activity	-	-	1	4.3	2	8.7	20	87	0.95	0.93
5.6 Evaluative questions	-	-	-	-	3	13	20	87	1.00	1.00
5.7 References for use	-	-	2	8.7	5	21.7	16	69.6	0.91	0.86

* Caption: CVI: content validity index.

**Table 4 ijerph-19-05297-t004:** Profile of target audience judges (*n* = 45).

Variables	*n*	%
Sex		
Male	4	9
Female	41	91
Age		
20 to 30 years	9	20
31 to 40 years	17	38
41 to 50 years	14	31
51 years or more	5	11
Marital Status		
Married or living with a partner	29	65
Divorced	2	4
Separated	1	2
Widowed	-	-
Single	13	29
Race/color		
White	24	53
Brown	17	38
Black	3	7
Yellow	1	2
Indigenous	-	-
Education		
Middle school incomplete	1	2
Middle School complete	1	2
High school incomplete	2	4
High school complete	7	16
College incomplete	22	49
College complete	12	27
Did not attend school	-	-
Monthly income		
1 to 2 minimum wages	27	60
2 to 3 minimum wages	6	13
More than 3 times the minimum wage	12	27
Retired		
Yes	44	98
No	1	2
Smokes		
Yes	4	9
No	36	80
Used to smoke	5	11
Self-perception of health		
Poor	18	40
Regular	17	38
Good	10	22
Very good	-	-
Number of medications being taken		
None	6	13
1 to 2 medications	9	20
3 or more medications	30	67
Illnesses beyond fibromyalgia		
None	5	11
1 or 2 illnesses	32	71
3 or more illnesses	8	18

**Table 5 ijerph-19-05297-t005:** Judgments of the target audience judges (*n* = 45) on *Amigos de Fibro*.

Items	Judgment									
	Totally Inadequate		Partially Inadequate		Adequate		Totally Adequate		CVI *	Kappa
	*n*	%	*n*	%	*n*	%	*n*	%		
1. Objectives										
1.1 Care innovation potential for people with fibromyalgia	-	-	-	-	12	26.7	33	73.3	1.00	1.00
1.2 Clarity of the program’s objective	-	-	4	8.9	15	33.3	26	57.8	0.91	0.83
1.3 Relevance of the theoretical framework	-	-	4	8.9	16	35.6	25	55.6	0.91	0.83
1.4 Basic premises capable of guiding people with fibromyalgia in promoting their health	-	-	-	-	17	37.8	28	62.2	1.00	1.00
2. Proposed Themes and Initiatives										
2.1 The themes and initiatives listed offer support in promoting the health of people with fibromyalgia	-	-	-	-	14	31.1	31	68.9	1.00	1.00
2.2 The themes and initiatives listed are important to be applied to people with fibromyalgia in primary health care	-	-	-	-	16	35.6	29	64.4	1.00	1.00
3. Relevance										
3.1 Covers relevant topics for in promoting health for people with fibromyalgia	-	-	6	13.3	12	26.7	27	60	0.86	0.76
3.2 Addresses important premises for the practice of health education	-	-	5	11.1	18	40	22	48.9	0.88	0.79
3.3 Is innovative technology	-	-	-	-	23	51.1	22	48.9	1.00	1.00
3.4 Is a combination of multidisciplinary knowledge and practices	-	-	3	6.7	15	33.3	27	60	0.93	0.86
3.5 Has scalability potential	-	-	5	11.1	20	44.4	20	44.4	0.88	0.79
3.6 Is an approach that integrates multidisciplinary knowledge	-	-	-	-	18	40	27	60	1.00	1.00
4. Writing Style										
4.1 Ordering of content and initiatives	-	-	2	4.4	20	44.4	23	51.1	0.95	0.90
4.2 Clarity of the text	-	-	2	4.4	12	26.7	31	68.9	0.95	0.90
4.3 The vocabulary used	-	-	-	-	15	33.3	30	66.7	1.00	1.00
4.4 The extent of the material	-	-	-	-	18	40	27	60	1.00	1.00
4.5 Complexity of the text	-	-	-	-	17	37.8	28	62.2	1.00	1.00
4.6 Common vocabulary for professionals	-	-	2	4.4	17	37.8	26	62.2	0.95	0.90
5. Structure of the Program										
5.1 Number of participating professionals	-	-	-	-	21	46.7	24	53.3	1.00	1.00
5.2 Structure of the program (number, frequency, time/duration, period of the day, day of the week)	-	-	5	11.1	20	44.4	20	44.4	0.88	0.79
5.3 Conditions offered by the professional	-	-	2	4.4	23	51.1	20	44.4	0.95	0.90
5.4 Activities performed by the participant	-	-	2	4.4	20	44.4	23	51.1	0.95	0.90
5.5 Product of the participant’s activity	-	-	-	-	21	46.7	24	53.3	1.00	1.00
5.6 Evaluative questions	-	-	1	2.2	20	44.4	24	53.3	0.97	0.95
5.7 References for use	-	-	1	2.2	20	44.4	24	53.3	0.97	0.95

* Caption: CVI: content validity index.

## Data Availability

Not applicable.

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
