# Peer review of "Amigos de Fibro (Fibro Friends): Validation of an Educational Program to Promote Health in Fibromyalgia"

_ijerph, 2022, doi:10.3390/ijerph19095297_

Round 1

Reviewer 1 Report

In general, most sentences are too long and difficult to understand.

In addition, the reference style in the manuscript is not constant.

Especially, the description of the materials and methods/results sections is very confusing. Extensive revision must be needed.

"24" paragraphs are included in the discussion section.
It is difficult to understand the composition of an official paper.

Most sentences are too long and difficult to understand.  The current quality of the manuscript is too low.

Author Response

We would like to thank you for the considerations made in the manuscript “Amigos de Fibro (Fibro Friends): validation of an educational program to promote health in fibromyalgia” submitted to the International Journal of Environmental Research and Public Health and we are sure that the observations made greatly improved the quality. of the manuscript. Based on the adjustments made and highlighted in the file, we highlight the experts' questions and their respective answers.

Reviewer 2 Report

In this study, the authors validated a multidisciplinary educational health promotion program called Fibro Friends for individuals with Fibromyalgia in Brazil. The research was conducted in both expert judges and target audience. And the data was assessed by Content Validity Index (CVI) and Coefficient Kappa. Before the manuscript was suitable for publication, some issues needed addressing.

  1. A figure should be added to address the protocol design of the study.
  2. Why Fibro Friends was investigated in this study? Was there any other similar health education program? If yes, what were the advantages of Fibro Friends, compared to other(s) ?
  3. In Lines 94-96, the formula needed updating. The current one was confusing.
    4.CVI has two forms, including Item-CVI (I-CVI) and Scale-level CVI (S-CVI). Which one was discussed in this study? 5. Was other indicator analyzed in this methodological research besides Content Validity Index (CVI) and Coefficient Kappa?

Author Response

(The authors gave the same response as above.)

Reviewer 3 Report

This paper evaluates an educational health promotion program for fibromyalgia in the form of a questionnaire. Questionnaires were taken from professionals and patients, and the results are highly reliable because they were collated. An inherent bias of questionnaires is the tendency for favorable responses to be given, so the reader should be aware of this when reading the paper. Since few papers on similar studies have been reported, this paper is appropriate for publication because it is thought to contribute to advancing medical care for fibromyalgia. Since the paper is a report of the questionnaire survey results, there are no areas that require revision in terms of content. The only point that needs to be revised is the discussion that COVID-19 was not reflected in the paper because it occurred during the survey (L403-L408), but the author should add the discussion by citing other references.

Minor Correction

The beginnings are disconnected in the leftmost rows of Tables 2 and 4. The numbering should be left-aligned. If there is any significance to the different starting locations, it should be clearly stated in a footnote.

Author Response

(The authors gave the same response as above.)

Round 2

Reviewer 1 Report

No comments.

Author Response

Dear Editor and Reviewers,

We would like to thank you for the considerations made in the manuscript “Amigos de Fibro (Fibro Friends): validation of an educational program to promote health in fibromyalgia” submitted to the International Journal of Environmental Research and Public Health and we are sure that the observations made greatly improved the quality of the manuscript. We used the file already with the adjustments from the first round and highlighted the new adjustments that were mentioned by the two reviewers.

Reviewer 2 Report

The authors did not fully understand my comment #1. They need to describe how they designed this study. In other words, the author needed to add in a flowchart of their study. However, the current figure 1 (should be revised as table, not figure) could be attached as supporting information. 

Author Response

(The authors gave the same response as above.)
